# Perceived Barriers and Enablers for Taking Iron–Folic Acid Supplementation Regularly Among Adolescent Girls in Indonesia: A Pilot Study

**DOI:** 10.3390/ijerph22020209

**Published:** 2025-02-01

**Authors:** Healthy Hidayanty, Nurzakiah Nurzakiah, Irmayanti Irmayanti, Yuliana Yuliana, Helmizar Helmizar, Yahya Yahya

**Affiliations:** 1Department of Nutrition Science, Faculty of Public Health, Hasanuddin University, Makassar 90245, Indonesia; nurzakiah@unhas.ac.id (N.N.); irmayanti@unhas.ac.id (I.I.); 2Department of Family Welfare Sciences, Faculty of Tourism and Hospitality, Universitas Negeri Padang, Padang 25131, Indonesia; yuliana@fpp.unp.ac.id; 3Department of Nutrition, Faculty of Public Health, Universitas Andalas of Padang, Padang 25129, Indonesia; helmizar@ph.unand.ac.id; 4Department of Anthropology, Faculty of Social and Political Sciences, Hasanuddin University, Makassar 90245, Indonesia; yahyakadir31@gmail.com

**Keywords:** barrier, enabler, IFA supplementation intake, social cognitive theory

## Abstract

Non-compliance with iron and folic acid (IFA) supplementation is a significant contribution to the high prevalence of anemia among adolescent girls in Indonesia. This pilot study aims to explore the perceived barriers and enablers to regular IFA supplement consumption among adolescent girls. Using a qualitative approach, data were collected through focus group discussions (FGDs) conducted in Makassar and Padang City. The FGD involved 32 grade eight and nine students from four junior high schools. In-depth interviews were also conducted with health workers, teachers, and parents to triangulate the findings. A thematic analysis was performed using a social cognitive theory framework. The average age of the informants was 13.6 ± 0.6 years. Despite receiving IFA supplements at school, only 47% of informants reported consuming them regularly. Key barriers to regular supplementation included dislike of the taste and smell, parental prohibition, negative experience consuming IFA supplements, the belief that IFA supplements increase menstrual blood flow and volume, and forgetfulness factors. Enablers include self-awareness of the supplement’s benefits, trust in school-provided supplements, and positive support from parents and peers. The findings highlight that both barriers and enablers play a crucial role in influencing IFA supplementation adherence. Addressing these factors is essential for improving compliance and reducing anemia rates among adolescent girls. Given the small sample size and convenience sampling method, this study serves as a pilot, and further research is needed to validate these findings on a larger scale.

## 1. Introduction

Iron deficiency anemia (IDA) is one of the most prevalent nutritional deficiencies globally, especially affecting adolescent girls due to increased iron requirements during puberty. According to the World Health Organization (WHO), the prevalence of anemia among adolescent girls is notably high, with estimates indicating that approximately 27% of adolescent girls in developing countries are affected by this condition [1,2]. In the Asian context, the prevalence of IDA among adolescent girls is alarming. For instance, a study in Nepal found that anemia is particularly prevalent among adolescent girls, with significant implications for their health and future pregnancies [3]. In Indonesia, the situation is similarly concerning. A study in Indonesia found that the prevalence of iron deficiency anemia among adolescent girls was 21.1%, indicating a significant health challenge that requires targeted interventions [4]. This prevalence is exacerbated by factors such as menstruation, which leads to iron loss, and inadequate dietary iron intake, which is often observed in this demographic [3,5].

In order to address this public health concern, the WHO has implemented various programs. One significant initiative is the recommendation for weekly iron and folic acid supplementation (WIFAS) targeted at adolescent girls, particularly in regions where the prevalence of anemia is high [6]. Aligning with global health recommendations, the Ministry of Health of the Republic of Indonesia has implemented a program aimed at providing iron supplementation specifically for adolescent girls, which is crucial in addressing the high prevalence of anemia in this demographic. This initiative is encapsulated in the iron and folic acid (IFA) supplementation programs, which is outlined in the Circular Letter from the Ministry of Health Number HK.03.03/V/0595/2016. This document stipulates that adolescent girls and women of reproductive age are to receive one iron tablet weekly for a year, totaling at least 52 tablets annually. Adolescent girls received the majority of iron and folic acid (IFA) supplements in school [7]. The ministry’s initiative not only addresses the immediate health needs of adolescent girls but also aims to improve their overall nutritional status, thereby contributing to better health outcomes in the long term [8].

Despite the availability and distribution of these supplements, adherence rates remain suboptimal in many regions, posing a challenge to improving adolescent health outcomes. Globally, adherence rates to IFA supplementation programs among adolescent girls have shown more variability. A systematic review highlighted that adherence can range widely, with some studies reporting adherence rates as low as 9% in certain populations [9]. Based on the Indonesian National Report, adherence rates to IFA supplementation programs among adolescent girls have been very low. In 2018 and 2023, the compliance rates for IFA consumption were 1.4% and 3.0%, respectively [7,10]. Recent studies indicate that adherence rates vary significantly based on geographical and socio-economic contexts. For instance, a study conducted in East Kalimantan, Indonesia, revealed an alarmingly low adherence rate of only 1.4% among adolescent girls participating in the Weekly Iron Folic Acid Supplementation (WIFAS) program [11]. This low adherence is attributed to various factors, including lack of awareness, inadequate health education, and cultural beliefs surrounding supplementation [11].

The consumption of iron and folic acid (IFA) supplementation is crucial for adolescent girls, addressing immediate health concerns and providing long-term benefits. Research indicates that IFA supplementation significantly improves hemoglobin levels and reduces the prevalence of anemia among adolescent girls. For instance, a systematic review highlighted that IFA supplementation could lead to a 33% reduction in anemia risk within a few months [12]. Furthermore, studies conducted in various regions, including India and Ghana, have shown that school-based IFA supplementation programs effectively enhance hemoglobin levels and decrease anemia prevalence [13]. In addition to immediate health benefits, the consumption of IFA tablets has long-term implications for future pregnancies. Adequate iron and folic acid levels are essential for preventing complications during pregnancy, such as low birth weight and pre-term delivery [14]. Moreover, the intergenerational cycle of malnutrition can be broken by ensuring that adolescent girls receive adequate nutrition, thereby enhancing the health of future generations [15].

Despite the known benefits of IFA supplementation, adolescent girls often face various barriers to regularly consuming it. These barriers may include lack of awareness, misconceptions about its necessity, forgetfulness, and socio-cultural influences that affect adherence [16,17,18]. On the other hand, factors such as family support, proper counseling, and accessibility to supplements serve as enablers that can enhance adherence to IFA tablet consumption [19]. Therefore, understanding these barriers and enablers is crucial for developing effective interventions that increase adherence rates and improve health outcomes among adolescent girls.

Although several studies have explored the effectiveness of IFA supplementation programs, limited research has focused on understanding the specific barriers and enablers perceived by adolescent girls in different socio-cultural contexts. Previous studies have often focused on the clinical outcomes of iron deficiency without addressing the behavioral and psychosocial factors that affect adherence [20]. Moreover, research has primarily been conducted in adult women or pregnant populations, neglecting the unique needs and challenges faced by adolescent girls. The last five years have seen a growing recognition of the importance of tailoring health interventions to adolescent populations, but comprehensive studies addressing both barriers and enablers in diverse settings remain sparse [21].

Applying social cognitive theory (SCT) to examine the barriers and enablers of IFA tablet consumption among adolescent girls is crucial for designing more effective interventions. SCT posits that personal, behavioral, and environmental factors interact to influence health behaviors, making it a suitable framework for exploring adherence to supplementation among adolescents. One significant barrier identified in the literature is the lack of knowledge regarding the importance of iron and folic acid supplementation. For instance, a study by Waltengus et al. highlights that pregnant women with a good understanding of anemia were significantly more likely to adhere to supplementation [22]. This finding aligns with the principles of SCT, which emphasize the role of knowledge and self-efficacy in behavior change [23]. Moreover, environmental factors such as access to healthcare services and social support systems are critical enablers of adherence. For example, Nabugoomu et al. utilized SCT to explore how environmental influences, including community support and healthcare access, affect the nutritional behaviors of adolescent mothers [23].

While previous studies have explored the clinical efficacy of IFA supplementation, there remains a gap in understanding the psychosocial and environmental factors that influence adherence, particularly among adolescent girls. Much of the existing research has focused on adult populations, leaving a critical gap in the adolescent context [20]. Additionally, many studies have failed to incorporate behavioral theories such as SCT to explain why some girls adhere to supplementation programs while others do not. A comprehensive exploration of these factors, including personal beliefs, social influences, and environmental barriers, is necessary to enhance program effectiveness. Therefore, using SCT as a theoretical framework, this pilot study aimed to explore the barriers and enablers perceived by adolescent girls regarding regular IFA supplement consumption in the Indonesian context.

## 2. Materials and Methods

This pilot study employs a qualitative research design to explore the barriers and enablers perceived by adolescent girls regarding regular IFA supplementation consumption. It was conducted from May to July 2024 in Makassar City and Padang City, two urban areas that are represent the western and middle part of Indonesia, as well as represent the two kinship system, named patrilineal in Makassar City and matrilineal in Padang City. The initial hypothesis of the research was that the barriers and enablers differed from a cultural perspective in the two cities.

Data were collected using focus group discussions (FGDs), an approach that facilitates the exploration of participants’ perceptions, attitudes, and experiences regarding specific topics. The discussions were facilitated by the research team (H.H., and N.N.) from the background of health science, and the team had experience in conducting qualitative studies. They acted as a moderator who encouraged interaction among participants, allowing for the emergence of diverse perspectives that might not surface in one-on-one interviews [24,25]. A research assistant acted as a notetaker and noted down important things during the discussion.

A total of 32 adolescent girls were selected from four junior high schools that have implemented the IFA supplementation program from the two cities. Of this number, none of the participants refused to participate. Considering that there are several classes in grades 8 and 9, we performed class randomization using a random number generator. After obtaining the classes, we selected several informants who would be FGD participants. A purposive sampling technique was used to select informants with the following inclusion criteria: (1) had experienced monthly menstruation, (2) had obtained approval from the school principal to participate in this pilot study, and (3) was willing to participate in the pilot study after receiving the explanation of the pilot study objective. Given the pilot nature of this study, the sample size was intentionally small, and participants were selected through convenience sampling. This approach was used to gather initial insights that could inform future larger-scale studies on this topic.

Data were collected through six focus group discussions (FGDs), each consisting of 5–6 participants, and they were conducted in school without disturbing the school hour. The researcher’s first meeting with the informant was when the FGD was to be conducted at school. Before the discussion, the researcher introduced themselves to participants and explained the objective of the pilot study. Participants were asked to join the pilot study on a voluntary basis by filling in and signing an informed consent form. The discussion was conducted in a classroom where there was no one else except for the informants, and they sat in a circle. Each informant was asked to respond to the question freely and confirmed if there were any unclear questions from the moderator. The moderator asked for confirmation for any unclear answers from the informant to ensure the accuracy of the informant’s response. Following the discussions, we transcribed the data and conducted interactive coding for preliminary analysis. Recruitment of informants and discussion were concluded once no new information emerging during the discussion process. The discussions were conducted in Bahasa Indonesia (the national language of Indonesian) and lasted between 60 and 90 min. All FGDs were audio-recorded and later transcribed verbatim.

Discussions were guided using a developed semi-structured interview. The draft guide was reviewed by a multidisciplinary team and sent to the expert team to check the guide. After receiving feedback from the expert team, the draft guide was revised, and the final version was then translated to Bahasa Indonesia. The guide was also piloted on female students at a junior high school that had similar characteristics to the study site to assess their understanding of the guide. The final version was then conducted by using a researcher team to guide the interview. Key topics discussed included factors influencing adolescents’ adherence or non-adherence to regular IFA supplementation, parental and other family attitudes towards supplements, experiences with side effects, and the perceived benefits of regular IFA supplementation.

An in-depth interview was performed to validate the information raised from FGDs as a triangulation of information. Informants for in-depth interviews were health workers who handled the IFA supplementation program for schools, parents who encouraged and prohibited their daughters to consume IFA, and teachers in charge of the IFA program at school. We conducted interviews with two health workers, two parents, and two teachers. The first contact with the informants was when visiting their workplace, the community health center (Puskesmas), home, and school for the conducted interview. Before meeting the informants, we made an appointment with them by phone. And when we met, we explained the purpose of the pilot study, asking for their consent to participate and arranging a time to conduct an interview. Before conducting interviews, we asked for permission to record the interview. In order to keep confidentially of the informant during interviews and avoid contamination of information from others, we conducted interviews in the personal room, where there was no other interference. Interviews were conducted by a research team who had experience in performed in-depth interview. After being interviewed, informants were given a reward for the time given for the interview. All interviews were audio-recorded and later transcribed verbatim.

The information collected from health workers was the implementation mechanisms of the IFA supplementation program, whether counseling was carried out on the importance of consuming IFA and the possible side effects experienced by students after consuming IFA, and monitoring related to student compliance in consuming IFA regularly. The data collected from parents were the reasons why they encouraged and prohibited their daughters from consuming IFA regularly. The data collected from teachers were the roles they played in relation to IFA supplementation in their schools.

The study report followed the standards for reporting qualitative research [26]. Data analysis followed the principles of thematic analysis. All transcribed data were checked. If there was local dialect or jargon, it was then translated to Bahasa. This process was performed by two researchers (H.H. and N.N.) and double-checked by a research assistant who understood the jargon to avoid mistranslation. The final transcripts were coded manually by having all of the transcripts systematically read by one researcher (I). Then, the codes categorized responses into themes that reflected both barriers and enablers of IFA supplementation consumption. The further analysis was performed using NVivo version 12. Themes were identified inductively, allowing the data to guide the analysis without imposing pre-conceived categories. Thematic coding was guided by the SCT framework, focusing on factors such as self-efficacy, outcome expectations, and social support. Themes were discussed regularly by the researchers H.H., N.N. and Y.Y., through which they reached final themes. Information from the in-depth interviews and observation of participants’ expression were also summarized.

Confirmability, credibility, dependability, and transferability were used to enhance the trustworthiness of the data. Data checking from researchers’ notes and data collection in the form of triangulation of information were employed to achieve confirmability. To assure credibility, the interview guide and protocol were piloted, as well as peer debriefing. The discussions were audiotape-recorded to minimize bias and ensure the correctness of capturing information from discussions and also to support credibility. The detail of pilot study protocol was prepared to ensure dependability. The details of the data collection and analysis help to compare with other studies to ensure transferability.

Ethical approval for this pilot study was obtained from the relevant institutional review boards (IRBs) in Makassar City. Prior to data collection, informed consent was obtained from the participants and their parents. The participants were assured of confidentiality, and pseudonyms were used in the transcripts and reports. Participation in the pilot study was voluntary, and the girls were informed that they could withdraw at any time without any consequences [27]. All recordings and transcripts were securely stored, and access was restricted to the research team.

## 3. Results

### 3.1. Characteristic of Informants

Informants for the FGD consisted of 32 adolescents’ girls. The average biological age of the 32 total informants was 13.6 ± 0.6 years. All informants already had regular monthly menstruation since they were 11 years old (43.8%). Most of the girls’ fathers were businessmen (88%), and most of their mothers were housewives (59%). Of the total informants, there was 46.9% that did not consume IFA supplementation regularly.

The analysis of the focus group discussions (FGDs) revealed several key themes regarding the barriers and enablers to IFA supplementation among adolescent girls in Indonesia. These themes were categorized into three main areas: perceived barriers, perceived enablers, and perceived need for IFA supplementation consumption (Table 1).

### 3.2. Perceived Barriers to Regular IFA Supplementation Consumption

Several barriers to regular IFA supplementation consumption were identified. The most prominent barriers included:1.Taste and smell: As many as 8 out of 15 informants who did not routinely consume IFA reported that they disliked the taste and smell of the IFA supplementation, which made them reluctant to consume it regularly. This sensory aversion was a common theme across the informants. Below are the quotes from informants to support the explanation of these barriers.

“I don’t like…the taste of blood”.(S.A., 14 years)

“I once consume IFA supplementation and then vomited, I don’t want to do it again”.(M.S., 14 years)

“I only took it occasionally. Because I don’t like the smell, it makes me nauseous, and after consuming the IFA supplementation, I often feel dizzy”. (F.A., 13 years)

Conversely, among the participants who regularly consumed IFA supplementation, one informant reported experiencing nausea due to its taste. However, this adverse reaction did not deter her from continued consumption; instead, she employed alternative strategies to mitigate the discomfort. Below are the quotes from the informant to support the explanation of this barrier.

“For instance, after consuming this supplement at school today, I typically experience nausea and headaches due to its odor. I then take a brief rest, and upon waking, I feel more energized and motivated”. (L.O., 13 years)

Information from health workers stated that the provision of iron tablets to junior high school students has been carried out since 2016 based on the decision of the Indonesian Minister of Health. The provision of IFA is carried out once a month according to the number of students in each school. The allocation for one student is 4 IFA tablets, which are given through the school health service teacher. The provision of IFA is accompanied by the provision of education about IFA, which includes what anemia is, the impact of anemia, the benefits of consuming IFA, and the side effects of consuming IFA. Education is only carried out in the new school year and also if there is a request from the school. The following is an excerpt from an interview with health workers.

“During nutrition education, we provide an explanation about anemia, the impact of anemia both now and in the future, the importance of consuming IFA supplementation, also the possible side effects of IFA supplementation, and the importance of consuming food before consuming IFA to reduce the side effects of consuming it such as nausea”. (M.W., HW1)

“Education about anemia and IFA supplementation was implemented during the new school year. Duration of education was delivered in five to ten minutes and held in school field. In addition, education was given if there is a request from the school. Usually based on monitoring from the UKS teacher, if compliance of IFA consumption is low, so we are asked to provide education again”. (S.L., HW2)

2.Parental and family influence: Informants mentioned that their parents prohibited them from taking the tablets (6 participants out of 15 informants who did not routinely consume IFA supplementation), often due to concerns about potential side effects or misconceptions regarding this supplementation. Below are the quotes from informants to support the explanation of these barriers.

“Even though we know the benefits, if the parents say no, we won’t take it. For example, I was supposed to get vaccinated at school, but I was prohibited from getting the vaccine”. (S.Q.,13 years)

“Actually, my family doesn’t recommend take supplement like that, but they recommend preventing eating random food. Bring supplies from home. It’s better like that than consuming supplement like that. That’s what my family often says”. (M.N., 13 years)

Information from students is supported by statements from parents who do not allow their children to consume tablets or medicines given at school. Parents prefer to encourage their children to consume food rather than take medicine because they feel that their children are not sick, so they do not need to consume IFA supplements.

“She is not sick, so there is no need to take medicine like IFA. I asked my child to bring food from home and improve her diet so that she is always healthy and can study well, no need to take medicine if she is not sick”. (I.S., PR1)

3.Previous negative experiences: Informants (6 participants out of 15 informants who did not routinely consume IFA) also shared previous negative experiences with supplements, such as gastrointestinal discomfort or side effects, which discouraged them from continuing with IFA supplementation. Below are the quotes from informants to support the explanation of these barriers.

“Sometimes, after taking IFA tablets, my friends suddenly feel dizzy. But when asked, they said they hadn’t eaten yet. However, there are also those who have eaten but still complain of dizziness, feeling like they want to sleep”. (M.U., 13 years)

“I took it once out of curiosity about the taste”. (K.I., 13 years)

“I used to get sick when I took the medicine that was distributed by the school, so since then I have not been allowed to take the medicine given by the school”. (F.R., 14 years)

4.Perception of increased menstrual blood flow: There was a widespread belief among the participants that consuming IFA supplementation would lead to increased blood volume during menstruation, which further deterred them from taking the tablets regularly, especially when they are menstruating (7 of 15 informants who did not routinely consume of IFA supplementation). Below are the quotes from informants to support the explanation of these barriers.

“Because I was worried that a lot would come out due to the many school activities lasting until the afternoon, I was afraid of leaking”. (U.R., 12 years)

“Indeed, more blood comes out, it hurts and it’s uncomfortable at school”. (S.A., 13 years)

In addition, there was also one informant who believed that consuming IFA supplementation actually makes menstrual frequency irregular.

“I feel that if I take IFA my period becomes irregular and doesn’t run smoothly, for example this month on the 11th, sometimes next month I don’t have my period at all”. (T.S., 14 years)

5.Forgetfulness: Of the 15 informants who did not take IFA supplementation regularly, a total of 11 informants stated that the reason was because they forgot. IFA supplementation distribution is carried out routinely at school, and it is expected to be drunk immediately after being given. However, some did not drink it immediately, but kept it in their pockets and would consume it during breaks or at home. In reality, they forgot, or the supplement was scattered because of its small size. Below are the quotes from informants to support the explanation of these barriers.

“I also take this supplement. However, sometimes I forget or it slips from my pocket”. (N.K., 13 years)

“Sometimes I also take this supplement, if I remember to take it”. (T.S., 14 years)

“I don’t take the supplements given by school regularly because uh… I forget. The schedule is quite busy, you know”.(G.K., 13 years)

### 3.3. Perceived Enablers to Regular IFA Supplementation Consumption

Several enablers were identified that encouraged the participants to take IFA supplements. Below describes the enablers as well as quotes from informants to support the results.

1.Self-awareness of health benefits: Of the total informants, there were 21 informants that expressed awareness of the health benefits they experienced after consuming the supplements, such as improved energy levels and general well-being, which motivated them to continue the supplementation.

“I usually feel dizzy when I first wake up. But after taking this supplement, I don’t feel as dizzy”. (P.U., 14 years)

“Yeah, same here. It’s like it maybe gives more energy… increases blood too, right? Like, it decreases. So, taking the supplements helps increase our blood. That way, we don’t lack it and don’t get anemia”. (F.A., 13 years)

“To prevent anemia… to boost energy… so that we have the strength”. (S.I., 15 years)

“Even though I don’t take this supplement regularly, I know that it is beneficial for my body”. (G.K., 13 years)

2.Trust in school-based programs: Participants had a positive perception of the IFA supplementation programs administered through schools (12 of the total informants). They expressed trust in the information and guidance provided by the school health staff, which acted as a significant enabler.

“I belief that what are given to us from school must be good for us. So, I consumed this supplement”. (S.Q., 13 years)

“Regularly reminding by teacher to consume this supplement”. (T.S., 14 years)

Information provided by teachers reveals that every month, health workers from the community health center come to the school. They bring the supplements to the school. Then, IFA supplementation is distributed to female students by the UKS (*usaha kesehatan sekolah*/school health services) teacher.

“This program collaboration with UKS teacher and teacher class. Every Friday, we distribute IFA supplementation to female students and ask students to take IFA supplementation”. (A.S., SP1)

“In Friday we called *jumat berkah*, where student have lunch together. We asked the female students to consume of IFA supplementation because it is can support them in study”. (B.B., SP2)

3.Parental and peer support: The role of social support, particularly from parents and peers, was highlighted as a critical enabler in encouraging adherence to IFA supplementation by a total of 12 informants. Positive reinforcement and shared experiences among peers played an important role in adherence to IFA supplementation.

“It’s like I was afraid to take just anything. But when my friends explained about the IFA supplementation, I thought, ‘Oh, it’s okay,’ as long as it’s clear what it is and its purpose”. (A.U., 13 years)

“Uh…My mom told me, ‘Take the IFA tablets so you don’t get anemia’. Then she said, ‘Fafa usually has low blood pressure, so try taking it’”. (F.A., 13 years)

“My father reminds me to take my IFA supplementation. He said ‘Mutia don’t forget to take your supplementation’. That’s why I never forget to take it”. (M.U., 13 years)

This result supported by information from parent that urge in taking IFA supplementation.

“I know that it is important for adolescent girls consume IFA supplementation…to prevent anemia. Therefore, I asked my child to consume it which received from school”. (H.M., PR2)

### 3.4. Perceived Need to Regular IFA Supplementation Consumption

In addition to the barriers and enablers expressed by total informants, they also revealed what they needed to reduce the barriers they faced and to strengthen the enablers in order to support regular IFA supplement consumption behavior, especially support from themselves (12 informants), parents and family (6 informants), friends (6 informants), teachers (4 informants), and health workers (4 informants).

“I had to strengthen myself by saying, ‘If I don’t take this supplement, I will definitely feel dizzy’”. (S.A., 13 years)

“If my parents told me to, I would definitely take this supplement. However, even if I wanted to, if my parents did not support me, I would not take this supplement”. (S.Q., 13 years)

“My parents remind me every day, telling me to take the IFA supplementation because sometimes I forget to take it”. (R.A.,14 years)

“I hope that teacher reminded the students through words at school every few days to take IFA supplementation”. (T.S., 14 years)

“The presence of friends is important…to remind me”. (N.A., 14 years)

“Health workers come to schools to provide information about anemia, using interesting media”. (Z.K., 13 years)

## 4. Discussion

The findings of this pilot study offer valuable initial insights into the barriers, enablers, and perceived needs related to IFA supplementation consumption among adolescent girls in Indonesia, framed within social cognitive theory (SCT). As a pilot study, these results are exploratory and aim to guide future research by identifying key themes. SCT, which suggests that personal, behavioral, and environmental factors interact to shape health behaviors, was an appropriate framework for this initial exploration of supplementation adherence. The personal barriers identified, such as aversion to the taste and smell of IFA (leading to nausea and gastrointestinal discomfort), previous negative experiences, perceptions of increased menstrual blood flow, and forgetfulness, were prominent themes. These barriers not only reduce adolescents’ willingness to adhere to IFA supplementation but also highlight the influence of broader misconceptions and cultural beliefs surrounding health and menstruation. However, given the pilot nature of this study, further research with larger samples is needed to validate these findings and assess their generalizability.

Adolescents often experience nausea and gastrointestinal (GI) discomfort after taking iron and folic acid (IFA) supplements. This phenomenon can be attributed to several factors, including the pharmacological properties of the supplements, dietary habits, and individual physiological responses. The side effects of IFA supplementation are well documented. High doses of folic acid can lead to a range of gastrointestinal issues, including nausea, vomiting, and abdominal cramps, especially when combined with iron supplements [28]. The combination of iron and folic acid is known to exacerbate these side effects, leading to increased reports of nausea and vomiting among adolescents [28]. Additionally, the unpleasant taste and smell of iron tablets can contribute to non-compliance and aversion to the supplements, further complicating adherence to supplementation regimens [11].

Dietary factors also play a significant role in the experience of nausea and GI discomfort. Adolescents with poor eating habits, such as low meal frequency and inadequate dietary diversity, may be more susceptible to the side effects of IFA supplements [29]. For instance, consuming the supplements on an empty stomach can heighten the likelihood of nausea and vomiting [30]. Furthermore, the presence of dietary inhibitors of iron absorption, such as phytates found in certain grains and legumes, can lead to gastrointestinal distress when combined with iron supplements [31].

To mitigate these adverse effects, several strategies can be employed. It is recommended that adolescents take IFA supplements with food to reduce the incidence of nausea and other GI symptoms [30]. Additionally, educating adolescents about the importance of dietary diversity and the consumption of iron-rich foods can enhance the effectiveness of supplementation and reduce the risk of anemia [31]. Incorporating iron absorption enhancers, such as vitamin C-rich foods, can also improve the overall efficacy of iron supplementation [31].

In this pilot study, the provision of IFA supplementation to adolescents was carried out at school every Friday. On this day, students were asked to bring breakfast supplies to school and then have breakfast together in class; then, the teacher distributed IFA supplementation to students and asked them to drink on the spot so that the teacher could ensure that students consumed the iron tablets. However, there were still students who did not consume the iron tablets provided.

Negative experiences can create a psychological barrier, where the fear of discomfort discourages continued use of the supplements. Previous negative experiences with supplementation can lead to a general aversion to taking IFA. This aversion is compounded by a lack of adequate counseling and education regarding the importance of IFA supplements and the management of side effects [32]. Effective communication from healthcare providers about the benefits of IFA and strategies to mitigate side effects could enhance compliance. For instance, providing information on taking the supplements with food to reduce gastrointestinal discomfort may encourage adherence.

Moreover, perception of increased menstrual blood flow is one of the barriers that was found from this pilot study. This belief can deter adherence to supplementation programs, as many adolescent girls fear that these supplements could exacerbate menstrual symptoms or lead to heavier bleeding. Studies have indicated that misconceptions about the side effects of IFA supplements, including fears of increased menstrual blood flow, are prevalent among young women, leading to non-compliance with supplementation regimens [13].

One significant contributor to increased menstrual blood volume is the hormonal fluctuations that occur during the menstrual cycle. Prostaglandins (PGs), which are hormone-like substances produced in the endometrium, play a critical role in regulating menstrual flow. Elevated levels of PGs can lead to increased uterine contractions, which facilitate the expulsion of menstrual blood from the uterine cavity. Studies have shown that adolescents with dysmenorrhea, or painful menstruation, often exhibit higher levels of PGs, correlating with increased menstrual flow and duration of bleeding [33]. Another factor influencing menstrual blood volume is the age at menarche and the subsequent establishment of the menstrual cycle. Research indicates that many adolescents experience irregular cycles during the initial years post-menarche, which can lead to variations in menstrual blood volume. For instance, studies have reported that a significant percentage of adolescent girls experience prolonged menstrual flow, with some reporting cycles lasting more than seven days [34].

In this pilot study, education about anemia, including the importance of consuming IFA and managing its potential side effects, was provided by health workers from the local health center during a nutrition education session. However, this education was only conducted once at the start of the school year. The less-than-ideal delivery conditions and the brief duration of the session likely limited the students’ ability to fully comprehend the information. These factors could explain the students’ insufficient understanding of the importance of IFA supplementation, which may have contributed to the perception of side effects as barriers and may have resulted in lower compliance. As this is a pilot study, these findings highlight areas that could be improved in future, larger-scale interventions.

Forgetfulness can stem from a lack of routine or reminders, which is particularly prevalent among adolescents who may have busy schedules filled with school and extracurricular activities. Research indicates that forgetfulness is frequently cited as a primary reason for non-adherence to IFA supplementation among this demographic [35]. For instance, a study in Ethiopia highlighted that forgetfulness, alongside fear of side effects, significantly contributed to the nonadherence of pregnant women to IFA supplementation [36]. This suggests that similar patterns may exist among adolescent girls, who may not prioritize or remember to take their supplements regularly. To address the issue of forgetfulness, various strategies can be implemented. Educational interventions that inform adolescent girls about the importance of IFA supplementation and its benefits can enhance adherence. Additionally, integrating reminders into daily routines, such as using mobile applications or setting alarms, can serve as effective tools to combat forgetfulness.

Environmental factors such as parental and family influence are critical barriers of adherence. Research indicates that parental support significantly impacts adolescents’ adherence to supplementation programs. For instance, a systematic review highlighted that interventions aimed at improving adolescent nutrition often involve family engagement, which can enhance compliance with supplementation [16]. The involvement of parents not only provides emotional support but also facilitates the establishment of healthy dietary practices at home, which is essential for reinforcing the benefits of iron and folic acid intake [37]. Furthermore, studies have shown that adolescents who receive adequate health education about the importance of these supplements, often facilitated by their parents, are more likely to adhere to supplementation regimens [37]. This pilot study was conducted in two different cultural locations. In Indonesia, cultural norms surrounding reproductive health can create barriers for adolescents, particularly in patrilineal societies where discussing sexual health is often stigmatized [38]. This lack of open communication can lead to inadequate knowledge about reproductive health services, further exacerbating health issues among adolescents [39]. Conversely, matrilineal systems may facilitate more open discussions about health, leading to better health-seeking behaviors among adolescents [40].

Moreover, the family environment plays a critical role in shaping adolescents’ attitudes towards health behaviors. For example, a study conducted in Ethiopia found that family size and parental education levels were significantly associated with adherence to iron and folic acid supplementation among pregnant women, suggesting that similar dynamics could be at play for adolescents [41]. This aligns with SCT, which emphasizes the importance of observational learning and modeling behaviors within family settings. When parents prioritize and model healthy behaviors, adolescents are more likely to adopt similar practices [42].

According to this pilot study, parents can both encourage and discourage their teenagers from consuming IFA. Parental knowledge is the primary cause. Children are encouraged to take IFA by parents who recognize the significance of preventing anemia in teenagers, but children are not encouraged to consume IFA by parents who do not recognize the significance of preventing anemia. The fact that the original hypothesis about the importance of patrilineal and matrilineal societies in this pilot study was the same in both cities is another significant factor pertaining to parents. In both cities, mothers and fathers have an equal impact on their children’s health. In both Padang and Makassar, there is a similar level of communication with mothers and fathers about IFA supplementation. Teenage females are willing to talk.

The personal and environmental factors identified as enablers from this pilot study were self-awareness of health benefits, trust in school-based programs, and parental and peer support. The regular consumption of iron and folic acid (IFA) supplementation among adolescent girls is significantly influenced by self-awareness of health benefits and trust in school-based programs. Evidence suggests that when adolescent girls are aware of the positive impacts of IFA supplementation on their health, they are more likely to adhere to supplementation regimens. For instance, studies have shown that when adolescent girls are educated about the consequences of anemia and the role of IFA in preventing it, their compliance with supplementation increases markedly [43]. This self-awareness can be fostered through educational interventions that highlight the importance of maintaining adequate hemoglobin levels for overall health and cognitive function [44].

Trust in school-based programs is a critical enabler for the successful implementation of IFA supplementation initiatives. Evidence from various studies demonstrates that school-based programs not only facilitate the distribution of supplements but also enhance the perceived reliability of these interventions among students and their family [13]. In addition, in Ghana, a school-based program demonstrated that when teachers were involved in the distribution and education about IFA supplements, compliance rates improved significantly [45]. This suggests that leveraging trusted figures within schools can enhance the perceived credibility of health interventions, thereby increasing adherence among adolescent girls.

Furthermore, the role of social influences, such as peer support and community approval, has been emphasized in the context of adolescent health behaviors. This pilot study also found that adolescents need support from peer support, teachers, and health workers. Compaoré et al. found that community perceptions significantly impacted adherence to iron and folic acid supplementation among adolescents in rural Burkina Faso, illustrating the importance of social norms and peer influence as outlined in SCT [46]. This social dimension is crucial, as adolescents are often influenced by their peers and community, which can either facilitate or hinder their health-related decisions.

The integration of SCT in exploring the barriers and enablers for adolescents adhering to iron and folic acid supplementation underscores the pivotal role of parental and familial support. By fostering an environment that promotes health education and positive reinforcement, families can significantly enhance adherence to supplementation programs, ultimately contributing to better health outcomes for adolescents.

### Strengths and Limitations

This pilot study has several strengths, including its contextual relevance to addressing anemia among adolescent girls in Indonesia and its use of the social cognitive theory (SCT) framework to understand the factors influencing adherence to IFA supplementation. These strengths allow for the design of culturally appropriate interventions and inform public health strategies.

However, there are limitations. The focus on specific barriers may have overlooked broader factors, such as government policies or access to supplements in remote areas. The sample, limited to two cities (Makassar and Padang) and four junior high schools, may not fully represent the diverse adolescent population in Indonesia. While the sampling strategy targeted participants with relevant characteristics, future studies should include a larger, more geographically diverse sample to enhance generalizability. Additionally, the use of convenience sampling may introduce selection bias, which could affect the representativeness of the findings. Future research should consider randomized or stratified sampling to address these biases.

## 5. Conclusions

In conclusion, as part of this pilot study, several key barriers and enablers to IFA supplementation consumption among adolescent girls in Indonesia were identified. Although the pilot study’s sample size was small and findings should be interpreted with caution, the results provide important initial insights. The most prominent barriers identified among adolescent girls were related to the sensory properties of the supplements, parental influence, misconceptions about the effects of IFA supplementation on menstruation, and forgetfulness. On the other hand, enablers included self-awareness of health benefits, trust in school-based programs, and positive peer and parental support. Addressing these barriers and strengthening enablers are critical for future research and efforts to enhance IFA supplementation consumption and reduce anemia rates in adolescent girls.

Based on the findings, the following recommendations are proposed:Strengthening the role of health workers: Further research is needed to test the effectiveness of regular educational programs that clarify the benefits of IFA supplementation and address concerns about its effects. Initial findings suggest that health education may play a pivotal role in reshaping perceptions and improving compliance, but this should be confirmed in larger studies.Parental involvement: Preliminary results highlight the importance of developing educational programs targeting both adolescents and their parents to address misconceptions about IFA supplementation. More extensive research is required to assess the impact of correcting misinformation on improving adherence.School-based interventions: Schools appear to play a significant role in IFA supplementation programs, as trust in these institutions was identified as an important enabler. Further studies are needed to explore how health education in schools can be strengthened to foster positive attitudes towards supplementation.Peer support networks: Establishing peer support groups may increase adherence to IFA supplementation among adolescents. However, before widespread implementation, further research is necessary to evaluate the effectiveness and mechanisms of such interventions.

## Figures and Tables

**Table 1 ijerph-22-00209-t001:** Sub-sub-themes, sub-themes, and themes that emerged from the adolescents.

Sub-Sub-Themes	Sub-Themes	Themes
Bad taste	Taste and smell	Perceived barriers to regular IFA supplementation consumption
Do not like the smell		
Parental prohibition	Parental and family influence	
Family advice for maintaining health besides IFA supplementation		
Experience of nausea	Previous negative experiences	
Experience of vomiting		
Experience of dizziness		
Experience of headaches		
Leakage during school activities	Perception of increased menstrual blood flow	
Uncomfortable at school		
Irregular period of menstruation		
Inconsistently taking the supplement	Forgetfulness to consume IFA	
Busy schedule at school		
Improve energy	Self-awareness of health benefits	Perceived enablers to regular IFA supplementation consumption
Benefit to general well-being		
Prevent anemia		
Increased blood		
Trust in school program	Trust in school-based program	
Teacher reminding		
Support from father	Parental and peer support	
Support from mother		
Positive experience from peer		
Self-Strengthening	Self and social support	Perceived need to regular IFA supplementation consumption
Parents and family support		
Positive peer support		
Regular reminding from teachers		
Education from health workers		

## Data Availability

The data cannot be publicly available in repositories because they contain personal information.

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
