# Peer review of "Perceived Barriers and Enablers for Taking Iron–Folic Acid Supplementation Regularly Among Adolescent Girls in Indonesia: A Pilot Study"

_ijerph, 2025, doi:10.3390/ijerph22020209_

Round 1

Reviewer 1 Report

Comments and Suggestions for Authors

This is a useful small study of the attitudes of adolescent girls toward iron folic acid supplements in Indonesia. It is not clear if the group is representative of a larger group but nevertheless there is useful information, some of which needs further discussion in the discussion section. For example, do the supplements really taste or smell bad? If so, is there a way to correct that?  Where did the idea come from about the manuscript still needs editing

\\

Specifics

Round all numbers like age to only one place after decimal

Give the whole MS an English edit; punctuation is not in line with usual conventions

Are iron folic tablets called iron tablets only in Indonesia or elsewhere too?

Line 41 only very severe iron deficiency is likely to lead to impaired cognitive function etc. Not just a bit of anemia

Line 44 Are you sure ref 5 says later complications? This seems a bit overinterpreted.

Lien 68 and cultural beliefs about what? Iron supplements?

Lien 69 what is meant by self-regulation strategies; do you mean habit?

Lines 82 onward This looks like a convenience sample with some efforts to select using certain criteria (what were they) . The basic point in the discussion is that a limitation of the study is that it may not be representative of the population of adolescent girls in Indonesia since each island differs a lot as I remember and there was no effort to do a national sample, but there was an effort to get some  broad perspectives. Nevertheless only 32 kids aren’t very much. Yet   the results are interesting. This is really a pilot study

Line 125 and all of them were menstruating. Most of the girls’ fathers were businessmen and their mothers were housewives. This tables could be made into a narrative paragraph since there are only 32 participants

It would be good to point out numbers of subjects citing each barrier and if there were some who cited multiple barriers. Do the tables taste bad? Line 137.  If so, discuss later how it could be changed. Discussion should also focus on other factors that are probably true (Gi discomfort if the supplement was high dose) , parental misconceptions, etc., Then there are those that are truly misconceptions like dizziness, “energy” increased blood flow in menses etc. Discussion should focus on what to do about these misconceptions of girls and parents.

Overall, this is an interesting paper. It needs a limitations section, but it is useful even if a small pilot study upon which to base later work. The take away points are that the iron /folic tablets have problems which need fixing by manufacturers or distributors and there are also problems of communication that need fixing. Nice little piece of work. But a pilot and needs generalization to ensure that the factors are not just the function of the beliefs of a small number of girls. What about very poor kids who are most at risk? . 

Reviewer 2 Report

Comments and Suggestions for Authors

Although the topic is relevant, the research manuscript represents an analysis that does not significantly add any new element compared to existing studies. Many of the barriers and enablers discussed are already widely and well documented.

Apart of mentioned limitation, the use of only 32 participants, although with qualitative data, could limit the research outcome. The sample may not be representative of all Indonesian adolescents or other socio-cultural contexts.

Although focus groups approach is efficient, it would have been interesting to include quantitative data to reinforce the conclusions and verify the extent of the problem with a larger population.

Despite the use of social cognitive theory, the analysis seems to remain superficial. Further exploration of psychological and social factors would have been useful for understanding the dynamics that influence.  

Comments on the Quality of English Language

Minor editing of English is needed. 

Reviewer 3 Report

Comments and Suggestions for Authors

Dear Authors,

thanks for the study which aims to explore the barriers and enablers perceived by adolescent girls regarding regular iron supplement consumption.

Manuscript has significant flaws that prevent it from being published.

1. The introduction should be improved with the statistics data (it is not enough with data of WHO) and studies on the effects of iron deficiency on health.

2. The introduction lacks the aim of study.

3. The study lacks a hypothesis, which must be stated in the introduction.

4. Only 32 participants participate in the study, which is very small number of participants considering that 4 schools were included in the study.

5. The authors did not specify the criteria for the inclusion and exclusion of participants in the study.

6. There is only one participant in the study aged 15 years who should be excluded from the study.

7. There is insufficient data analysis in the results sections. Data should be analyzed according to age, menarche, religion, father occupation, mother occupation, and ethnic, otherwise it is not clear why this information about the participants is given.

8. The discussion section is underdeveloped. If a complete data analysis has been performed in the result section, then the authors will be able to expand the discussion accordingly.

9. Limitations of the study should be included in the discussion section.

10. Conclusions need to be revised/improved.

11. The first paragraph of conclusions repeats what is stated in the results and discussion sections.

12.  Recommendations 1 and 2 are for whom, who will do it?

13. References are insufficient.

Round 2

Reviewer 2 Report

Comments and Suggestions for Authors

The revised manuscript still misses critical points that I highlighted earlier. The lack of novelty remains a key concern, as the contribution is not significant. Additionally, the qualitative findings lack innovation. This manuscript is more appropriate for a regional or local journal rather than one aimed at a broader audience. As I previously mentioned, the limited number of participants represents a significant limitation. Therefore, in my opinion, it would be more suitable for publication in a scientific popular journal.

Comments on the Quality of English Language

Moderate editing is needed. 

Author Response

Dear Reviewer,

Thank you very much for taking the time to re-review this manuscript. We really appreciated for your valuable and meaningful comment and suggestions. Please find the detailed

responses below and the corresponding revisions/corrections highlighted/in track changes in

the re-submitted files. 

Comments 1 : The lack of novelty remains a key concern, as the contribution is not significant

Response 1 : Thank you for pointing this out. We agree with this point. Therefore we have added this information on discussion session line 434 - 439, 468 - 475, 514 - 523. 

Comments 2 : The qualitative findings lack innovation

Response 2 : Thank you for your comment on this point. We agree with this point. Actually in this study we also conducted in-dept interview among stakeholder (health workers, teachers, and parents) to support result from FGD . Therefore we add explanation on this in method section, line 150 - 158 and describe the result in the result section, in line 212-234, 251 - 250, 342 - 353, 370 - 375.

Comments 3 : The limited number of participants represents a significant limitation

Response 3 : Thank you for pointing this out. We realize that this study has involved limited number of participant. We mention it in out limitation part as well as strategy to minimize effect to result due to limited number of participants in line 567-572.

Reviewer 3 Report

Comments and Suggestions for Authors

Dear Authors,

thank you for improved manuscript, it has been significantly improved and now meets the requirements for a scientific publication.

Author Response

Dear Reviewer,

Thank you very much for your valuable feedback and comments on our manuscript. They are very valuable and meaningful to us. 

Best Regards

Healthy Hidayanty